# Chemopreventive Effect of Cooked Chickpea on Colon Carcinogenesis Evolution in AOM/DSS-Induced Balb/c Mice

**DOI:** 10.3390/plants12122317

**Published:** 2023-06-14

**Authors:** María Stephanie Cid-Gallegos, Cristian Jiménez-Martínez, Xariss M. Sánchez-Chino, Eduardo Madrigal-Bujaidar, Verónica R. Vásquez-Garzón, Rafael Baltiérrez-Hoyos, Isela Álvarez-González

**Affiliations:** 1Departamento de Ingeniería Bioquímica, Escuela Nacional de Ciencias Biológicas, Instituto Politécnico Nacional, Unidad Profesional Adolfo López Mateos, Zacatenco, Av. Wilfrido Massieu Esq. Cda. Miguel Stampa S/N, Alcaldía Gustavo A. Madero, Mexico City 07738, Mexico; cid.stephanie@gmail.com; 2Catedra-CONAHCYT, Departamento de Salud, El Colegio de la Frontera Sur-Villahermosa, Tabasco 86280, Mexico; xsanchez@mail.ecosur.mx; 3Laboratorio de Genética, Escuela Nacional de Ciencias Biológicas, Instituto Politécnico Nacional, Unidad Profesional Adolfo López Mateos, Zacatenco, Av. Wilfrido Massieu Esq. Cda. Miguel Stampa S/N, Alcaldía Gustavo A. Madero, Mexico City 07738, Mexico; eduardo.madrigal@gmail.com; 4Catedra-CONAHCYT, Facultad de Medicina y Cirugía, Universidad Autónoma Benito Juárez de Oaxaca, Oaxaca de Juárez 68120, Mexico; veronicavasgar@gmail.com (V.R.V.-G.); rbaltierrez@hotmail.com (R.B.-H.)

**Keywords:** colon carcinogenesis, chemoprevention, cooked chickpea

## Abstract

Chickpeas are one of the most widely consumed legumes worldwide and they might prevent diseases such as cancer. Therefore, this study evaluates the chemopreventive effect of chickpea (*Cicer arietinum* L.) on the evolution of colon carcinogenesis induced with azoxymethane (AOM) and dextran sodium sulfate (DSS) in a mice model at 1, 7, and 14 weeks after induction. Accordingly, the expression of biomarkers—such as argyrophilic nucleolar organizing regions (AgNOR), cell proliferation nuclear antigen (PCNA), β-catenin, inducible nitric oxide synthase (iNOS), and cyclooxygenase-2 (COX-2)—was assessed in the colon of BALB/c mice fed diets supplemented with 10 and 20% cooked chickpea (CC). The results showed that a 20% CC diet significantly reduced tumors and biomarkers of proliferation and inflammation in AOM/DSS-induced colon cancer mice. Moreover, body weight loss decreased and the disease activity index (DAI) was lower than the positive control. Lastly, tumor reduction was more evident at week 7 in the groups fed a 20% CC diet. In conclusion, both diets (10% and 20% CC) exert a chemopreventive effect.

## 1. Introduction

Colon cancer (COC) had the third-highest incidence rate and the second-highest mortality rate worldwide in 2020. The Pan American Health Organization/World Health Organization (PAHO/WHO) predicts that if no measures are taken, the incidence of COC will have a 60% increase in the Americas by 2030 [1,2]. Cell proliferation represents an important factor in tumor growth; the overexpression of proliferation cell nuclear antigen (PCNA) [3] and β-catenin, a highly unstable cytoplasmic protein, are correlated with cell proliferation in COC. β-catenin stabilizes once phosphorylated. It translocates into the nucleus, binds to transcription factors, and induces cell proliferation [4]. Another important marker for cell proliferation is the nucleolar organizing regions (NORs)—loops of DNA that transcribe genes for ribosomal ribonucleic acid (RNA). NORs have a high affinity for silver in the argyrophilic nucleolar organizing regions (AgNORs), which are NOR-associated acidic proteins that can be used as pointers of cell multiplication [5]. Inflammation is a characteristic of most types of cancer [6] linked to the presence of the enzyme cyclooxygenase-2 (COX-2), expressed in response to inflammatory stimuli at specific sites [7,8]. Inducible nitric oxide synthase (iNOS) is also related to inflammation, involved in both nitric oxide (NO) production and angiogenesis, including endothelial cell proliferation, migration, differentiation, and interaction with the extracellular matrix. Therefore, COX-2 and iNOS are promising therapeutic targets for cancer treatment [9].

On the other hand, COC is closely related to the Western diet, which contributes to a chronic inflammatory state. In addition, an unhealthy lifestyle, ethnicity, race, and environmental factors (documented in studies with patients who have migrated) play an essential role in the etiology of this disease [10,11]. Analogously, evidence suggests that diet constitutes a critical factor in COC prevention, for instance, including dietary fiber [12,13], proteins, peptides [14,15], unsaturated fatty acids [16,17], and phytochemicals [18,19]. These compounds are mostly found in plant-origin foods, including legumes.

Chickpea seed is one of the most extensively used legumes for human consumption.

It contains proteins (18.3–25%); complex carbohydrates (54.60 to 60.40%); soluble fibers (1.23–1.38%); insoluble fiber (14.1–23.2%); lipids (1.12 to 6.8%); vitamins; minerals (1.94 to 2.41%); oligosaccharides (ciceritol and raffinose); phenolics such as formononetin, genistein, and flavonol kaempferol; isoflavones such as biochanin A; and phospholipids [20,21,22]. Due to this complexity, chickpea seed is a functional food, showing health benefits when consumed regularly, i.e., it can reduce risk factors such as overweight and adiposity; improve intestinal function, blood lipid profile, blood pressure, and inflammation biomarkers; and exert prebiotic effects [23,24,25]. We previously reported the effect of the consumption of cooked chickpea (CC) seed in an azoxymethane (AOM)/dextran sodium sulfate (DSS)-induced COC mice model at 21 weeks, obtaining positive results [26]. However, since carcinogenesis is a chronic disease, the present investigation aims to assess the effect of CC consumption at different time windows in a COC mice model induced with AOM—a chemical agent that can initiate cancer by alkylating DNA and facilitating a mismatch of nitrogenous bases—and DSS—a sulfated polysaccharide that causes epithelial damage by binding to medium-chain fatty acids in the colon [27,28]. 

## 2. Results

### 2.1. Body Weight

Table 1 shows the weight change of the six experimental groups at three evaluation times (1, 7, and 14 weeks). In the PC group, body weight was 29% lower after 14 weeks compared with the NC group. On the other hand, the animals fed 10 and 20% CC diets showed weight increases of 47 and 35% after 14 weeks compared with the PC group. Additionally, the groups fed CC diets only had 27 and 23% higher body weight than the PC group after 14 weeks. Additionally, the 10% CC diet group had a 4% higher weight than the NC group, while the 20% CC diet group had a 4% lower body weight than the NC group after 14 weeks. Lastly, the AOM/DSS + 10% CC and AOM/DSS + 20% CC groups showed a weight decrease of 10 and 13%, contrasting with the NC group after 14 weeks.

### 2.2. Macro- and Microscopic Evaluation

#### 2.2.1. Disease Activity Index (DAI)

Toxicity signs (diarrhea, visible blood in the stool, or soft stools with blood) or DAI were absent in the NC group as opposed to the PC group at the end of the 14-week experiment (Figure 1A). In contrast, the two groups administered with the carcinogens and fed the CC diets showed diarrhea and visible blood in the stool compared to the NC group. Conversely, the 10 and 20% CC groups showed no dissimilarities against the NC group at the end of the 14 weeks. The PC group manifested diarrhea and visible blood in the stool at weeks 2, 6, 8, 10, and 12. Furthermore, the AOM/DSS + 10% CC and AOM/DSS + 20% CC groups exhibited a decrease in DAI of 42 and 29% compared with the PC group at the end of the 14 weeks. Finally, the 10 and 20% CC groups displayed no toxicity signs compared with the PC group after 14 weeks of experimentation.

#### 2.2.2. Colon Weight to Length Ratio (Colon Wt/L Ratio)

Figure 1B shows the weight/length relationship in the three periods evaluated. THe colon Wt/L ratio was significantly lower (*p* < 0.05) in the NC group than in the PC group in all three evaluations. The AOM/DSS + 10% CC group had an increase in colon Wt/L ratio compared with the NC group at the end of weeks 1 (43%), 7 (67%), and 14 (54%). Likewise, the colon Wt/L ratio in the AOM/DSS + 20% CC group was 39, 59, and 45% higher than the NC group in the three periods evaluated. On the other hand, the PC group had an increase in colon Wt/L ratio at weeks 1 (58%), 7 (72%), and 14 (64%) compared with the NC group. The group administered with AOM/DSS + 10% CC presented a 10, 3, and 7% decrease in colon Wt/L ratio in all three evaluations contrasted with the PC group. The AOM/DSS +20% CC group had a decrease in colon Wt/L ratio of 12, 8, and 12% at weeks 1, 7, and 14, respectively, compared with the PC group.

#### 2.2.3. Number of Aberrant Crypt Foci (ACF)

Figure 1C shows the number of ACF or preneoplastic lesions found in the colon of the analyzed groups. The ACF was eight times higher in the PC group than the baseline or the NC group in weeks 1, 7, and 14. Similarly, the AOM/DSS + 10% CC group exhibited 7-, 4-, and 6-fold increases in ACF against the NC group in the three periods evaluated. Likewise, the AFC was six, three, and three times higher in the AOM/DSS + 20% CC group than the NC mice in weeks 1, 7, and 14, respectively. Additionally, in the group treated with AOM/DSS + 10% CC, ACF decreased by 14, 43, and 28% compared with the PC mice after 1, 7, and 14 weeks. Moreover, AFC decreased by 29, 56, and 54% in the AOM/DSS + 20% CC group compared with the PC group in weeks 1, 7, and 14. The 10 and 20% CC groups showed lower ACF compared with the PC mice.

#### 2.2.4. Tumor Incidence

The incidence of tumors was observed macroscopically in weeks 7 and 14. No tumors or neoplastic lesions were found in the NC and 10 and 20% CC groups. The PC mice displayed a 100% incidence of tumors in both weeks 7 and 14. The groups AOM/DSS + 10% CC and AOM/DSS + 20% CC presented a decrease of 54 and 46% tumor incidence in week 7, and 85% and 77% after 14 weeks, contrasting with the PC group (Figure 1D).

#### 2.2.5. Histological Examination

Histological analysis of the colon in week 14 showed modifications in the histology between the carcinogen-induced groups and the diet-only groups (Figure 1E). In the PC group, 96% histological damage was observed compared to the NC group (Table 2). Moreover, the AOM/DSS + 10% CC and AOM/DSS + 20% CC groups had 40 and 45 % normal histology, contrasting with 100% in the NC group in week 14 of the evaluation (Table 2). Additionally, compared with the PC group, the AOM/DSS + 10% CC group had an 83 and 85% decrease in adenomas and adenocarcinomas. At the same time, the AOM/DSS +20% CC group had 85 and 91% fewer adenomas and adenocarcinomas than the PC group (Table 2). Moreover, the groups 10 and 20% CC exhibited no histological damage compared with the PC mice. On the other hand, Figure 1E indicates that the NC group had normal histology in the colon. Carcinogen-induced groups lost tissue structure, but in the groups AOM/DSS +10% CC and AOM/DSS +20% CC, the tissue structure had only minor changes.

### 2.3. Quantification of AgNOR, PCNA, β-Catenin, iNOS, and COX-2

#### 2.3.1. Proliferation Markers: AgNOR, PCNA, and β-Catenin

Immunohistochemistry was used to evaluate the proliferation markers AgNOR, PCNA, and β-catenin in colon tissue (Figure 2D). Figure 2A indicates that the PC group showed a 3- to 4-fold increase in the percentage of AgNOR expression contrasted with the NC group in week 3 of evaluation. Correspondingly, the AOM/DSS + 10% CC and the AOM/DSS + 20% CC groups showed a 2- to 3-fold increase in AgNOR expression compared with the NC group in weeks 1, 7, and 14. In addition, AgNOR expression decreased by 28, 28, and 22% in the AOM/DSS + 10% CC group and by 32, 33, and 31% in the AOM/DSS + 20% CC group in comparison with the PC group in weeks 1, 7, and 14. Lastly, the expression of AgNOR was lower in the diet-only groups (10% CC and 20% CC) than in the PC group.

Figure 2B shows that the PC group had 3-, 6-, and 5-fold increases in the PCNA expression in weeks 1, 7, and 14 of the experiment compared with the NC group. On the one hand, the expression of PCNA in the AOM/DSS + 10% CC group was three, five, and four times greater than in the NC group in weeks 1, 7, and 14. Additionally, the AOM/DSS + 20% CC group had three, six, and five times greater expression than the NC group after 1, 7 and 14 weeks of the experiment. On the other hand, PCNA expression decreased by 45, 53 and 60% in the AOM/DSS + 10% CC group and by 52, 45, and 52% in the AOM/DSS + 20% CC group compared with the PC group at the times of evaluation. Diet-only groups exhibited lower expression of PCNA than the PC group in weeks 1, 7, and 14 of the experiment.

Finally, Figure 2C shows the quantification of nuclei positive for β-catenin. Only the week 7 and week 14 results are presented on account of positive nuclei manifestation. The β-catenin expression was 13 times higher in the PC group than in the NC group in weeks 7 and 14 of the analysis. Likewise, the expression of this biomarker was 11 and 9 times higher in the AOM/DSS + 10% CC group than in the NC group in weeks 7 and 14. Furthermore, AOM/DSS + 20% CC showed an 8-fold increase in β-catenin expression by comparison with the NC group in both evaluation weeks. Additionally, β-catenin expression decreased by 21 and 24% in the AOM/DSS + 10% CC group and by 37 and 39% in the AOM/DSS + 20% CC group, both contrasted with the PC group in weeks 7 and 14. 

#### 2.3.2. Inflammation Markers: iNOS and COX-2 

Immunohistochemistry was used to evaluate the inflammation biomarkers iNOS and COX-2 in colon tissue (Figure 3C). iNOS expression (Figure 3A) was four to five times higher in the PC group than in the NC group in weeks 1, 7, and 14 of evaluation. Correspondingly, the treatment groups AOM/DSS + 10% CC and AOM/DSS + 20% CC had a 1- to 2-fold increase in iNOS expression contrasted with the NC group in weeks 1, 7, and 14. In comparison with the PC group in weeks 1, 7, and 14, the AOM/DSS + 10% CC group showed a reduction in iNOS expression by 42, 71, and 33% and the AOM/DSS + 20% CC group by 36, 60, and 48%. Lastly, the iNOS expression in both the 10 and 20% CC groups was significantly different (*p* < 0.05) from the PC group in weeks 1, 7, and 14 of the experiment.

Figure 3B indicates the percentage of expression of COX-2 in all groups. The expression of this biomarker was two to three times greater in the PC group than in the NC group at the evaluation times. Moreover, AOM/DSS + 10% CC had 2-, 3-, and 1-fold increases in COX-2 expression compared with the NC group after1, 7, and 14 weeks. Similarly, the AOM/DSS + 20% CC group showed a one to two times increase in the marker expression contrasted with the NC group at the evaluations. Analogously, COX-2 expression decreased by 21, 12, and 31% in the AOM/DSS + 10% CC group and by 17, 37, and 45% in the AOM/DSS + 20% CC group, both compared with the PC group in weeks 1, 7, and 14.

## 3. Discussion

Murillo et al. [29] reported similar results to this study regarding weight gain, describing that body weight had no significant changes in 15-week-old mice fed diets supplemented with chickpea flour (ideal weight 30–35 g for ICR strain). In addition, Cuellar-Nuñez et al. [30] observed a constant weight increase in ICR strain mice that were part of the negative control, where the animals had an increase of 50% in week 13 compared with week 1. Studies show evidence for a body weight loss between 5 and 20% in AOM/DSS-induced colon cancer mice [31], similar to the results observed in the PC group of this study. Moreover, Sanchez-Chino et al. [26] described that the weight gain of the animals was simultaneous in all groups except in those that were administered with AOM and DSS, where weight loss was observed from week 10 onwards. In patients with advanced cancer, cachexia usually occurs through two proteolytic signaling pathways activated by p-Stat3 in the muscle, causing muscle wasting or loss [32]. Silva et al. [32] found that cancer stimulates p-Stat3 in the muscle, triggering protein loss by stimulating caspase-3, myostatin, and the ubiquitin-proteasome system in a model of cancer cachexia carried out in 8- to 10-week-old female CDF2F1 mice. 

Furthermore, studies have utilized the DAI score to determine the severity of the damage caused by DSS by evaluating clinical signs (diarrhea and bleeding) in experimental animals [33]. Shi et al. [34] reported bleeding and soft stools one week after the administration of the DSS cycle and increased toxic signs in weeks 10 and 16 of the experiment. Tanaka et al. [8] also observed the presence of diarrhea and bleeding in the second week after induction, similar to the results from this work. The administration of DSS causes toxic symptoms such as diarrhea and bleeding (or their combination) since this compound causes inflammation and ulceration in the colon. Notwithstanding these outcomes, Monk et al. [35] determined that a diet supplemented with 20% CC for three weeks promoted the integrity of the epithelial barrier and improved intestinal health in mice, potentially because chickpea contains proteins, carbohydrates, lipids, and functional compounds (phenolics, saponins, and trypsin inhibitors, among others) that promote health.

Additionally, the colon Wt/L ratio is a marker of mucosal hyperplasia and the severity of chronic colitis [31]. Accordingly, Ju et al. [36] determined that the carcinogens employed in the AOM/DSS-induced colon carcinogenesis model in C57BL/6 mice shortened the colon length, caused edema, changed morphology, and induced tissue stiffness, making the colon heavier. Consistent with these results, Elimrani et al. [31] demonstrated that mice induced with AOM/DSS showed a reduction in colon length and increased weight at the end of the trial, indicating that AOM/DSS causes colon tissue stiffness and shortening. These authors [31] proposed that the colon Wt/L ratio increase in mice administered with AOM/DSS is related to a higher DAI and histological damage. 

Ju et al. [36] reported that a protective agent (bamboo salt) decreased the colon Wt/L ratio in AOM/DSS-induced colon carcinogenesis mice. In the present research, the colon Wt/L ratio diminished in the AOM/DSS-induced groups fed diets supplemented with 10 and 20% CC, but the diets did not show representative difference when compared with the PC group. 

On the other hand, ACF were found in all periods evaluated in this study but declined in number in the later evaluations due to the presence of neoplastic lesions. Murillo et al. [29] reported that the number of ACF or preneoplastic lesions decreased by 64% at the end of 10 weeks of an experiment with diets supplemented with 5% and 10% chickpea flour. Likewise, Sánchez-Chino et al. [15] evaluated the effect of chickpea protein hydrolysates on AOM-induced carcinogenesis mice fed hypercaloric and normocaloric diets at the end of 28 days, observing a 62% reduction in the number of ACF in the groups that consumed the hypercaloric diet. 

The carcinogenesis model used in this proposal contributed to the development of tumors in the latest evaluation periods (7 and 14 weeks of the study) as opposed to week 1—although AOM/DSS initiated the cells, the repair mechanisms were able to attenuate the damage and prevent tumor growth at that time. In the same way, Suzuki et al. [37] reported the appearance of tumors in the middle and distal parts of the colon tissue after ten weeks of AOM/DSS administration in mice, as shown in the results from the AOM/DSS-induced groups in this work. In addition, Bobe et al. [38] analyzed the chemopreventive effect of the ethanolic fraction of navy beans on AOM-induced colon carcinogenesis in obese mice, reporting a decrease in the incidence of any colon lesion compared with the group receiving the control diet due to the phenolic compounds contained in the ethanolic fraction. Similarly, Luna-Vital et al. [39] evaluated the antineoplastic potential of a peptide extracted from the indigestible fraction of common bean, declaring that it reduced inflammation and neoplasia formation, which suggests that the anticarcinogenic effect of the extract was a result of the synergism between the peptides in the extract. In the present research, nutritional and secondary metabolites from the whole chickpea seed could exert antineoplastic activity. 

Furthermore, Fleming et al. [40] proposed that testing biomarkers of proliferation and inflammation is crucial in the study of COC. The existing literature indicates a relationship between AgNOR expression as a marker of proliferation and COC patient survival, since the number of NORs in cancerous tissues is related to the percentage of cycling cells and the synthesis (S) phase of the cells [41]. Consistently, Ashokkumar and Sudhandiran [42] evaluated the protective effect of the flavonoid luteolin in a 17-week mouse model of colon carcinogenesis induced with AOM, describing that luteolin reduced cell proliferation, evidenced by lower AgNOR, PCNA, and β-catenin in contrast with the positive control. In this work, the expression of AgNOR decreased in the groups fed diets supplemented with 10% CC and 20% CC. However, the group receiving the diet with the higher percentage of supplementation showed a greater decrease in AgNOR expression. Moreover, Giron-Calle et al. [43] reported an 80% reduction in the proliferation of CaCo-2 colon cancer cells using a peptide fraction obtained from a chickpea concentrate applied for 4–6 days. Due to a direct relationship to cell proliferation, PCNA expression is a vital prognosis factor in colon carcinoma, which can be utilized as a marker of disease progression since it is related to high-grade dysplasia and increased colonic tissue size [44]. In line with this approach, Sanchez-Chino et al. [26] assessed the effect of diets supplemented with 2 and 10% CC on a mice AOM/DSS model, describing that PCNA was expressed in a lower percentage in the crypts of the animals that consumed a diet with 10% CC and reporting a 38% decrease in the overexpression of PCNA compared with the positive control group. On the other hand, Guajardo-Flores et al. [45] evaluated extracts of saponins and flavonols from germinated black beans to evaluate their antioxidant and antiproliferative activity in cancer cell lines, finding high antiproliferative activity attributed to the presence of quercetin. Additionally, Giron-Calle et al. [46] reported that peptides obtained from a chickpea protein hydrolysate could regulate cell proliferation in Caco2 and THP1 cell lines by 45% and 78%. 

Moreover, Magee et al. [47] analyzed the antiproliferative activity of chickpea-derived protease inhibitors in breast and prostate cancer cell lines, which reduced cell proliferation by 12–14 and 32–37%, respectively. Colon cancer is related to adenomatous polyposis coli (APC) inactivation and β-catenin alteration, since the Wnt/β-catenin signaling pathway enhances the repair of the nonhomologous end-joining of DNA double-strand breaks in cells [48]. Kumar et al. [49] evaluated the antiproliferative effect of black bean extracts on breast cancer cell lines, detecting anticancer properties through apoptosis induction and antiproliferative activities via cell cycle arrest in the S and G2/M phases in two cell lines. In addition, Feregrino-Pérez et al. [50] studied the chemopreventive effect of a polysaccharide (indigestible carbohydrates) extract from the common bean. Non-digestible carbohydrates are fermented in the colon and produce short-chain fatty acids, including butyrate, which can regulate cell proliferation genes (β-catenin), cell arrest, and apoptosis through the inhibition of the activity of histone deacetylase enzymes, which remove the acetyl groups from histones, preventing gene transcription by condensing the DNA structure and increasing the activity of suppressor gene tumors [51]. Xu and Chang [52] reported that hydrophilic extracts of chickpea flour had antiproliferative activity in colon cancer cells (SW480), with an IC_50_ of 5.71 mg/mL. Additionally, Sánchez-Chino et al. [26] declared an inhibition of β-catenin translocation to the nucleus when ICR mice were fed a 10% CC diet in an AOM/DSS model. 

The prolonged induction of iNOS during conditions of chronic inflammation leads to the formation of reactive NO intermediates, which are highly mutagenic and cause DNA damage, base mispairing, alterations in cell signaling, and the promotion of proinflammatory and angiogenic properties of the cell that ultimately contribute to carcinogenesis evolution [53]. Hence, Mazewski et al. [54] tested the anticancer activity of black lentil aqueous extracts in a mice model, observing a decrease in pro-inflammatory cytokines compared with the positive control group after 11 weeks of treatment. Comparably, Kim et al. [55] studied the protective effect of an ethanolic chickpea extract in a model of ulcerative colitis with DSS in mice, which suppressed the expression of pro-inflammatory interleukins, COX-2 and iNOS, potentially via the inactivation of NF-κB and activator of transcription 3 (STAT3). Likewise, Zhang et al. [56] analyzed the antioxidant and anti-inflammatory effects of nondigestible fermentable components and phenolic compounds of common beans in a model of acute colitis in mice fed a diet supplemented with 20% bean meal, concluding that such a diet reduced inflammation markers. Additionally, Arpon et al. [57] indicated that a diet based on proteins and polysaccharides from plants increases the total number of bifidobacteria, and short-chain fatty acids are associated with changes in the methylation patterns of genes related to inflammation.

COX-2 is frequently expressed in various types of cancer, in which it plays a multifaceted role in the genesis or promotion of carcinogenesis and the resistance of cancer cells to chemotherapy and radiotherapy [58]. Therefore, Boudjou et al. [59] analyzed two extracts of broad beans and lentils; the lentil extract had the greatest antioxidant and anti-inflammatory activity by inhibiting the COX-2 pathway. Additionally, Kim et al. [60] studied the anti-inflammatory effect of an isoflavone isolated from *Sophora japonica* L. (Leguminosae) on mice. This reduced paw edema and, consequently, the isolated isoflavone was identified as a selective inhibitor of COX-2. Lastly, Sánchez-Chino et al. [26] demonstrated that chickpea consumption suppressed iNOS and COX-2 expression in a model of colitis-associated colon cancer, as shown in the results of the present study.

Research attributes anti-colon cancer activity to CC compounds—saponins, protease inhibitors, bioactive peptides, phenolic compounds, indigestible fermentable carbohydrates, and short-chain fatty acids, such as butyrate—through antiproliferative, anti-inflammatory, and antioxidant properties. Hence, the synergism between these compounds may explain the anticancer activity of CC [26,43,46,47,55,61,62,63].

## 4. Materials and Methods

### 4.1. Chemicals

AOM (Azoxymethane) was purchased from Sigma–Aldrich (CAS 25843-45-2, St. Louis, MO, USA) and DSS (Dextran Sulfate Sodium (colitis-grade)) from MP Biomedicals (PM. 36,000–50,000 M W, CAS 9011-18-1, Irvine, CA, USA).

### 4.2. Chickpea (Cicer arietinum) Seed

Chickpea seeds (variety: Kabuli, color: cream, and crop year: 2016) were purchased at a local market in Mexico City. Later, they were soaked in distilled water at a 1:4 ratio (*w*/*v*) for 12 h and cooked in a pressure cooker with water (1:5 ratio) at 121 °C/25 min. Then, the seeds were drained, lyophilized (0.045 mbar/−45 °C/48 h, Labconco Lyophilizer, MO, USA), and subsequently ground into a fine powder (50 mesh, 0.297 mm). The chemical compositions of raw chickpea (RC) and CC were reported in Cid-Gallegos et al. [64] as follows: 7.11% and 9.76% lipids, 25.17% and 27.32% proteins, and 63.07% and 60.25% carbohydrates.

### 4.3. Animals, Induction of Colon Cancer, and Diets

The study protocol was approved by the ethics committee of the National School of Biological Sciences/National Polytechnic Institute (ENCB/IPN) (Approval No. CEI-ENCB-011-2017) on 14 June 2017. Male BALB/c mice, 6–8 weeks old, weighing 20–25 g, were obtained from the Autonomous University of the State of Hidalgo (Hidalgo, Mexico) research animal facility. The animals were housed under stable conditions (12 h light/dark cycles at 23 °C) for seven days, with standard food and drinking water *ad libitum*. The protective effect of CC on colon carcinogenesis (Figure 4) was evaluated based on the model proposed by Tanaka et al. [8]. Six groups of 7 mice were randomly selected for each period (1, 7, and 14 weeks). Colon cancer was induced in mice through the intraperitoneal injection of two doses of 10 mg/kg body weight AOM, one every five days, followed by administration of 1.5% DSS in drinking water for two cycles of five days, each with three rest days between each cycle. The animals were sacrificed after 1, 7, and 14 weeks of the experiment. Mice in group 1 were used as a negative control (NC) and fed standard food to observe the basal levels of both proliferation and inflammation markers. Group 2, or the positive control (PC), served to monitor the development of carcinogenesis and the modification of the selected markers. Groups 3 (AOM/DSS +10% CC) and 4 (AOM/DSS +20% CC) received the same treatment as PC but their diet was replaced with 10% or 20% CC 15 days before induction with AOM/DSS. Groups 5 (10% CC) and 6 (20% CC) were fed diets supplemented with 10 or 20% CC, respectively. Mice were euthanized by cervical dislocation at the end of 1, 7, and 14 weeks of the experiment. Afterward, the colon was removed and washed with cold (4 °C) PBS (pH 7.4) for subsequent analysis.

### 4.4. Toxicity Signs

Signs of toxicity were monitored in each mouse weekly, for instance, body weight and disease activity index (DAI) [30]. The DAI was calculated using a 4-point scale: 0 = normal, 1 = soft stools without visible blood (diarrhea), 2 = visible blood in the stool, and 3 = soft stools with blood [30].

### 4.5. Colon Weight/Length (mg/cm) Ratio

After euthanizing the animals, the colon was measured in weight and length to determine the weight/length ratio (colon Wt/L ratio) [36].

### 4.6. Number of Aberrant Crypt Foci (ACF)

Aberrant crypt foci were examined in the whole colon tissue from each animal in the experimental groups. After washing with PBS pH 7.4 (4 °C), colon tissue was placed in a Petri dish, embedded with solid paraffin, and fixed with 4% formaldehyde in PBS (pH 7.4) for 24 h. Afterward, each tissue was stained with 4% methylene blue in PBS for 2 min. Subsequently, the number of crypts was identified and imaged with an optical microscope (Carl Zeiss, Primo Star) at 10X magnification [65].

### 4.7. Tumor Incidence

The incidence of tumors in each mouse per group was calculated. Then, the percentage of tumors in each group was determined [31].

### 4.8. Histological Examination 

For histological analysis, the colon tissue previously fixed with 4% formaldehyde in PBS (pH 7.4) for 24 h was dehydrated in ethanol solutions at increasing concentrations (70%, 80%, 92%) and placed in a chloroform–xylol mixture (1:1). Later, the samples were embedded with paraffin (Paraplast, Leica). In addition, 5 µm sections of the embedded tissues were sliced with a microtome (Leica RM2125 RTS), mounted on a slide, and stained with hematoxylin & eosin (H&E). H&E-stained slides were observed and photographed under a light microscope (Carl Zeiss, Primo Star with an AxioCam ERc5s camera in combination with the ZEISS Labscope iPad app). Each micrograph was evaluated and classified according to Shi et al. [34] into four categories: normal, dysplasia, adenoma, and adenocarcinoma.

### 4.9. Quantification of AgNOR, PCNA, β-Catenin, iNOS, and COX-2

For histochemical analysis, the colon tissue was previously fixed with 4% formaldehyde in PBS (pH 7.4) for 24 h, dehydrated, and embedded with paraffin, as described in Section 4.8. Subsequently, (RM2125 RTS, Leica, Wetzlar, Germany) the embedded tissues were sliced into 4 µm sections with a microtome and placed on a slide. Argyrophilic nucleolar organizing regions (AgNORs) were stained with the silver technique, applying modifications to the thermal treatment (microwave), time (3 min), and temperature (55 °C). In addition, the concentration of the gelatin and the silver nitrate solutions were adjusted [66]. The slides were hydrated in xylol solutions and then in ethanol solutions at decreasing concentrations (96%, 90%, 80%, 70%) and submerged in a citrate buffer (pH 6). A 2% gelatin solution in distilled water was prepared and then mixed with formic acid for a final concentration of 0.5%. This was kept at 37 °C for 40 min. Additionally, a 25% silver nitrate solution was prepared in distilled water and centrifuged at 9000 rpm for 2 min at 23 °C. Both solutions were mixed at a 1:2 ratio. Thus, the slides were immersed and incubated at 37 °C for 20 min. After staining, the slides were washed three times with distilled water, dehydrated for 3 min in a Coplin jar with warm distilled water (60 °C), and mounted with resin.

Regarding the immunohistochemical analysis, the fixed colon tissue was dehydrated and embedded with paraffin, as mentioned in Section 4.8. Then, the embedded tissues were sliced into 3 µm sections using a microtome (RM2125 RTS, Leica) and placed on a slide with 4% 3-aminopropyl-trimethoxy-silane. Next, the expression of PCNA, β-catenin, INOS, and COX-2 was evaluated through immunohistochemistry. For PCNA, iNOS, and β-catenin slides were placed in a Coplin jar where the antigens were recovered in citrate buffer (pH 6) in a microwave oven (55 °C) for 4 min. After five washes with PBS (pH 7.4), 0.1% PBS-Triton was added to the slides for 5 min. Subsequently, endogenous peroxidase was reacted with 4% or 6% hydrogen peroxide (H_2_O_2_) in methanol for 30 min for β-catenin, PCNA, and iNOS, respectively. Additionally, the cells were blocked with 3% bovine serum albumin (BSA) in PBS for 1 h at room temperature. Then, the slides were incubated overnight at 4 °C using mouse monoclonal anti-PCNA (1:200, Thermo Fisher MA5-11358, Waltham, MA, USA), rabbit polyclonal anti-iNOS (1:200, ABCAM ab3523), and rabbit polyclonal anti-β-catenin (1:250, ABCAM ab2365, Cambridge, UK) antibodies diluted in PBS (pH 7.4). Later, they were washed with PBS (pH 7.4)−0.05% Tween and blocked with 3% BSA in PBS−0.05% Tween for 10 min at room temperature. Finally, secondary antibodies were added—goat anti-mouse IgG (Biotium 20400, Fremont, CA, USA) at 1:500 dilution for PCNA and goat anti-rabbit IgG (HRP) (Thermo Fisher 656120) at 1:200 dilution for iNOS and β-catenin—with 3% BSA in PBS and incubated for 90 min at room temperature.

For COX-2, the slides were placed in citrate buffer (pH 6), washed with PBS (pH 7.4), and 0.1% PBS-Triton was added for 5 min. Then, they were placed in 6% H_2_O_2_/methanol for 30 min and blocked with 3% BSA in TBS (TRIS base-NaCl (pH 7.6)) for 10 min at room temperature. Later, the slides were incubated overnight at 4 °C with rabbit polyclonal anti-COX-2 antibody (1:100, ABCAM ab15191) diluted in TBS (pH 7.6). Subsequently, they were washed with TBS (pH 7.6)-Tween 0.05% and blocked with 3% BSA in TBS-Tween 0.05% for 10 min at room temperature, the secondary antibody (goat anti-rabbit IgG (HRP), Thermo Fisher 656120) was added at a 1:200 dilution in 3% BSA in TBS and they were incubated for 90 min at room temperature.

The revelation was carried out using 3,3′-diaminobenzidine (DAB) as the chromogen and Harris hematoxylin (HYCEL, MX) as the counterstain. Next, tissues were dehydrated with distilled water (60 °C) for 3 min and mounted with resin (HYCEL, MX). Lastly, they were observed under an optical microscope and photographed with a 40X objective; 10 random fields were quantified for all groups with ImageJ 1.52p software (National Institute of Health, Bethesda, MD, USA).

### 4.10. Statistical Analyses

The results were summarized as mean ± standard error of the mean (SEM). One-way analysis of variance (ANOVA) with Dunnett’s or Tukey’s comparison tests were performed to identify significant differences (*p* ≤ 0.05) between groups.

## 5. Conclusions

A diet that prioritizes plants involves limiting the intake of animal-based foods and incorporating vegetable-based protein sources, such as legumes. Cooked chickpeas are an excellent food choice as they offer nutritional and biological benefits, including antioxidants, anti-inflammatory properties, and anti-cancer compounds that can help prevent chronic non-communicable diseases such as colon cancer. This study has shown that substituting 10–20% of the diet of the experimental animals, which were given carcinogens, with cooked chickpeas decreased macroscopic markers such as DAI, ACF, and tumor incidence over three testing periods. Furthermore, preneoplastic lesions were observed within the first week of AOM/DSS induction. With regard to the progress of colon carcinogenesis during the established periods (1, 7, and 14 weeks), the expression of macroscopic and microscopic markers was highest in week 7 for PC. This study observed that chickpea seeds have anti-proliferative and anti-inflammatory activity, making them a great addition to a plant-based diet for their protein, vitamin, mineral, and secondary metabolite content. Cooked chickpeas have also been shown to have anti-proliferative and anti-inflammatory activity in all three phases of colon carcinogenesis. This is due to the synergistic action of nutritional and non-nutritional compounds in cooked chickpeas. Future work prospects include evaluating the impact of chickpea consumption on other organs, pro-inflammatory and anti-inflammatory interleukins, and the effect during the cell cycle and cell death. Specific compounds or their combination should also be evaluated to compare the results with those obtained in this study.

## Figures and Tables

**Figure 1 plants-12-02317-f001:**
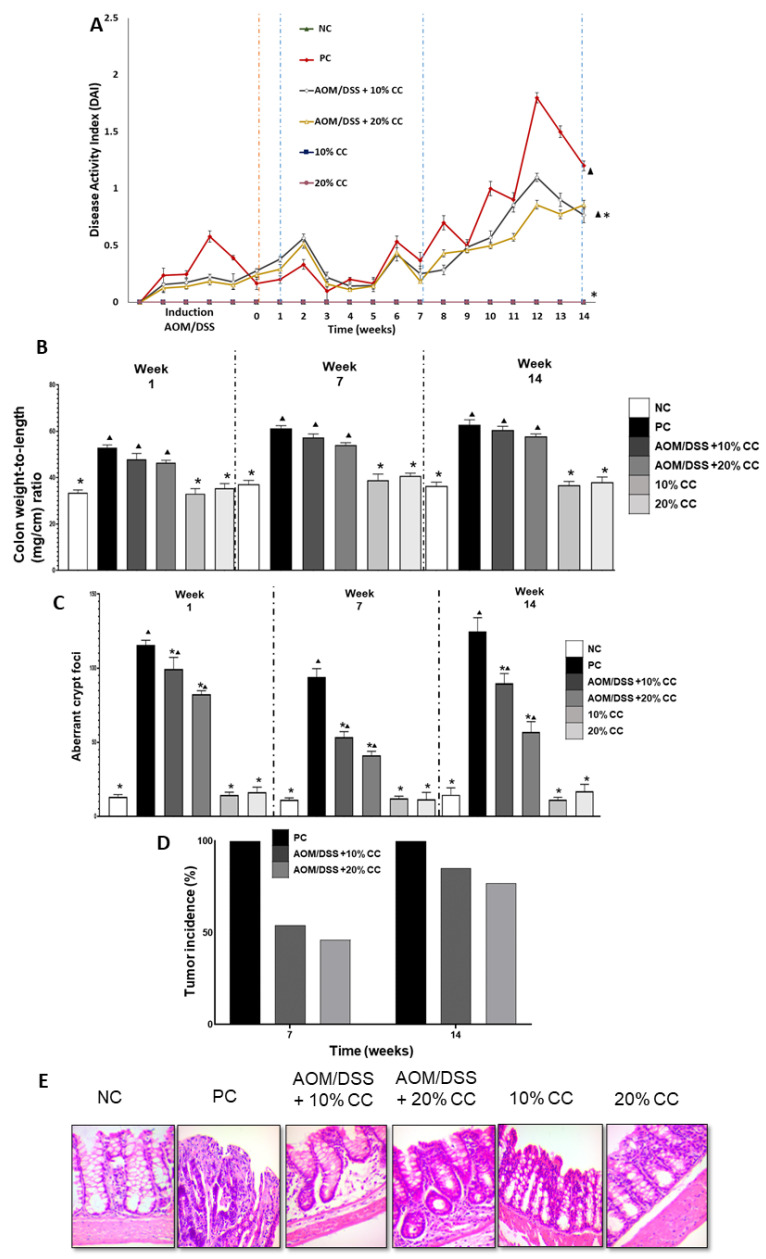
(**A**) Disease activity index over the 14 weeks of the experiment. (**B**) Colon weight to length (mg/cm) ratio at 1, 7, and 14 weeks of the experiment. (**C**) Number of aberrant crypt foci after 1, 7, and 14 weeks of the experiment. (**D**) Tumor incidence in the colon after 7 and 14 weeks of the experiment. (**E**) Images of colon histology H&E in week 14. Magnification 40×. * Significant difference compared with the PC group. ▲ Significant difference compared with the NC group. One-way ANOVA (*p* ≤ 0.05), Dunnett’s test.

**Figure 2 plants-12-02317-f002:**
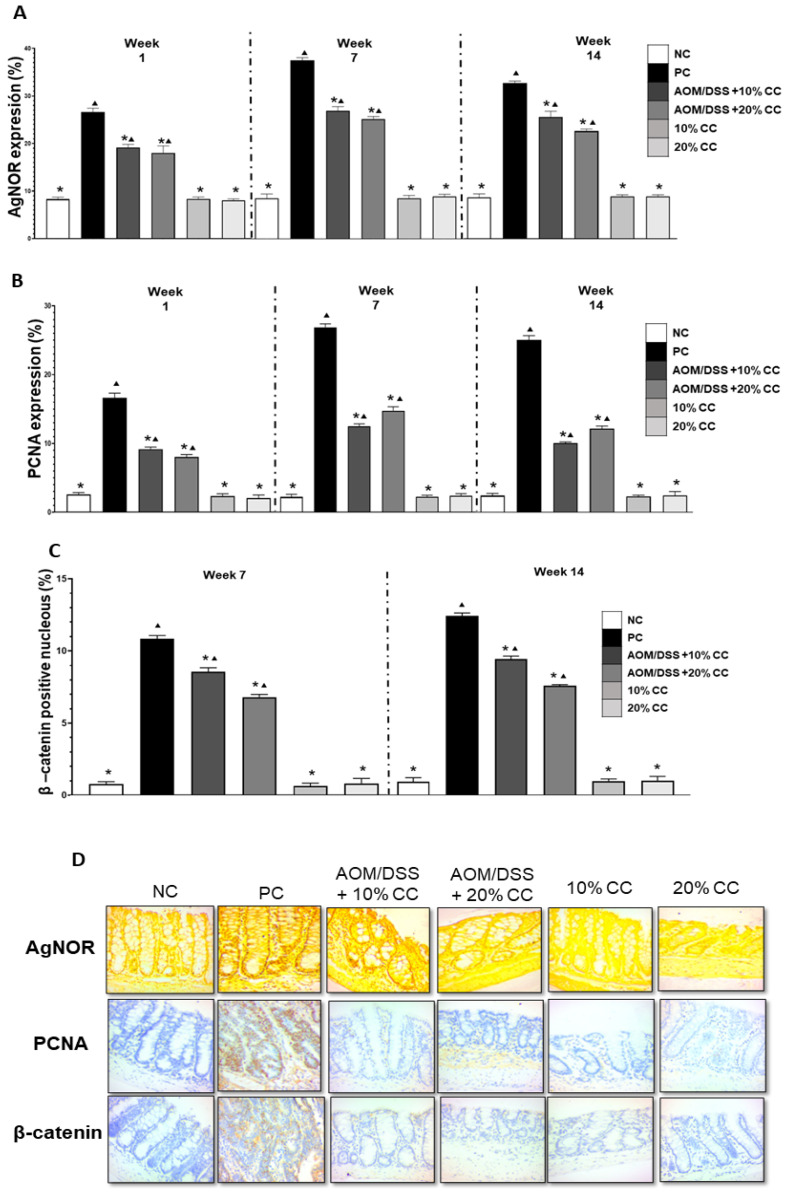
(**A**) Percentage of AgNOR expression in BALB/c mice colon at three evaluations. (**B**) Percentage of PCNA expression in BALB/c mice colon at three evaluations. (**C**) Percentage of β-catenin expression in BALB/c mice colon at two evaluations. (**D**) Images of colon tissue from immunohistochemistry (AgNOR, PCNA, and β-catenin) in week 7. Magnification 40×. * Significant difference compared with the PC group. ▲ Significant difference compared with the NC group. One-way ANOVA (*p* < 0.05), Dunnett’s test.

**Figure 3 plants-12-02317-f003:**
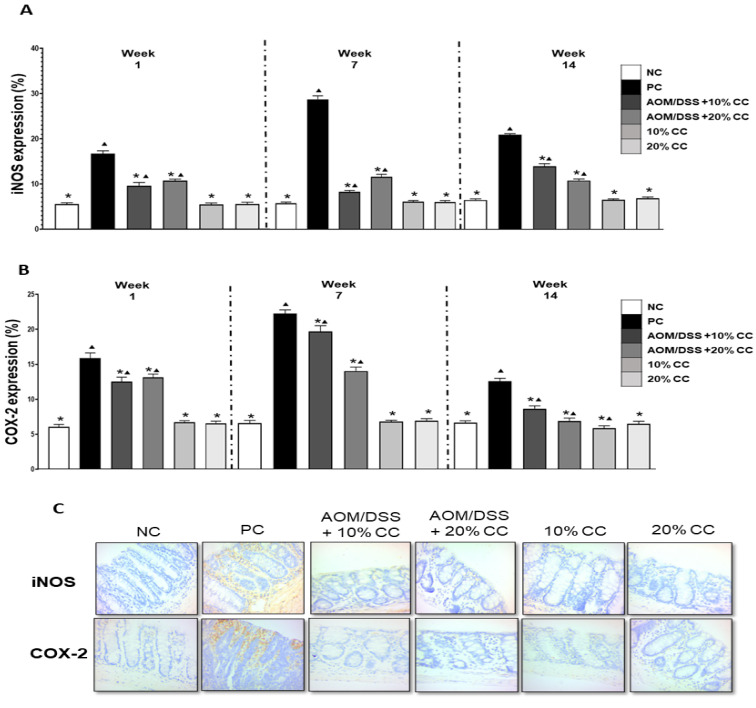
(**A**) Percentage of iNOS expression in BALB/c mice colon at the three evaluation times. (**B**) COX-2 expression in BALB/c mice colon at three evaluation times. (**C**) Images of colon tissue from immunohistochemistry (iNOS and COX-2) at week 7. Magnification 40×. * Significant difference compared with the PC group. ▲ Significant difference compared with the NC group. One-way ANOVA (*p* < 0.05), Dunnett’s test.

**Figure 4 plants-12-02317-f004:**
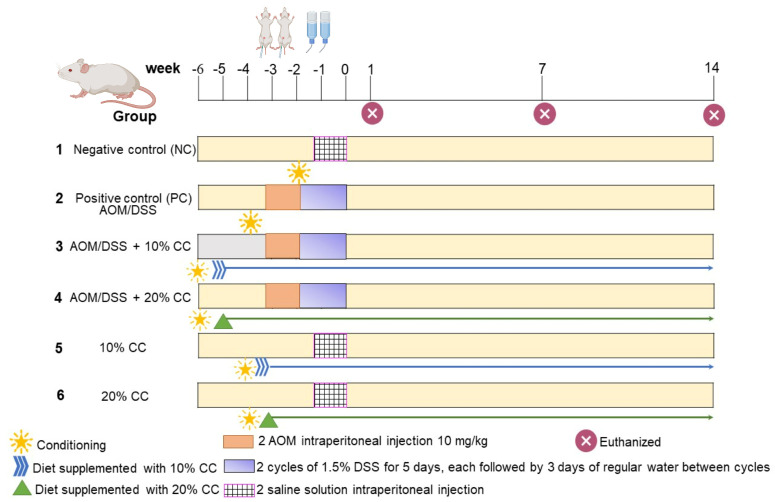
Schematic timeline of the AOM/DSS murine model for colon carcinogenesis and the administration of chickpea diets. Created in Biorender.

**Table 1 plants-12-02317-t001:** Body weight (g) at the end of 1, 7, and 14 weeks of treatment.

Treatment	Week 0	Week 1	Week 7	Week 14
	Body weight (g)			
NC	22.68 ± 0.31	25.40 ± 0.11	26.94 ± 0.25	28.9 ± 0.20 *
PC	22.51 ± 0.26	24.01 ± 0.30	26.13 ± 0.13	20.47 ± 0.46 ▲
AOM/DSS + 10% CC	22.36 ± 0.46	25.46 ± 0.65	25.32 ± 0.38	25.91 ± 0.20 *
AOM/DSS + 20% CC	22.55 ± 0.73	25.84 ± 0.61	25.70 ± 0.37	25.10 ± 0.48 *
10% CC	22.45 ± 0.23	26.55 ± 0.41	28.58 ± 0.18	30.10 ± 0.16 *
20% CC	22.61 ± 0.25	24.63 ± 0.44	26.71 ± 0.10	27.72 ± 0.25 *

Values represent the mean ± SE (*p* < 0.05). * Significant difference contrasted with the PC group at week 14. ▲ Significant difference contrasted with the NC group at week 14. One-way ANOVA (*p* < 0.05), Tukey’s test.

**Table 2 plants-12-02317-t002:** Histological damage evaluation of Balb/c mice colons after 14 weeks of the experiment.

		%			
Group	Treatment	Normal	Dysplasia	Adenoma	Adenocarcinoma
1	NC	100	0	0	0
2	PC	4.2 ± 3.7 ^a^	39.6 ± 3.2 ^a^	26.1 ± 8.1 ^c^	29.8 ± 12.1 ^c^
3	AOM + 10% CC	40.1 ± 4.0 ^b^	51 ± 5.0 ^c^	4.3 ± 5.0 ^b^	4.5 ± 4.7 ^b^
4	AOM + 20% CC	45.3 ± 4.0 ^c^	47.4 ± 1.3 ^b^	3.8 ± 2.8 ^a^	2.6 ± 1.8 ^a^
5	10% CC	100	0	0	0
6	20% CC	100	0	0	0

The results represent the mean ± SE (*p* < 0.05). Different letters in the columns represent significant differences between groups. One-way ANOVA (*p* < 0.05), Tukey’s test.

## Data Availability

Not applicable.

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
