# Peer review of "Chemopreventive Effect of Cooked Chickpea on Colon Carcinogenesis Evolution in AOM/DSS-Induced Balb/c Mice"

_plants, 2023, doi:10.3390/plants12122317_

Round 1
Reviewer 1 Report
No particular comment except that the authors state in the Statistical analyses (subsection 4.10) to use Duncan comparison test, but in the legends of Fig. 1 and 2 they refer to Dunnet's test.
Reviewer 2 Report
This paper is covering an interesting topic and the results contribute to the knowledge of the field. In general, the document contains many typos, grammar, and language errors. Discussion must improve.
*Introduction*
Lines 66-67: reference 12 does not support this information. This information is part of the introduction of reference 12, not a part of their results.
*Materials and methods*
Line 450- include the full name of the acronyms (AAOM and DSS)
Line 456- What is the variety? the color? crop year?
Line 457- what is the justification to eliminate the water?
Line 481- describe the group “PC”
*Results
Line 128- specify that you are referring to DAI
Section 2.2.1- instead to state that the value is a “significant difference”, indicate if it is significantly lower, higher, etc..
Line 144: which was higher? The group or the AFC number? Take the redaction.
Lines 166-169: describe the “damage” that is observed in each image.
Lines 170-174: include “Table 2”
Discussion*
This section has very long paragraphs making it difficult to read. The redaction must be improved as well as the paragraphs. In general, some parts of this section describe the results of other authors but the integration/connection to the results of the present research is very weak.
Lines 323-328: it is an example of something as an expected discussion. However, how the chemical composition of the chickpeas is compared to the phenolic content of Bobe? Another way, what is the reason to mention it? Many cases as it can be found in the discussion section.
Lines 263-265- Include information about the ideal weight of the animals at the mentioned age.
Lines 217-290: ¿ Do the animals of the present research show bleeding and diarrhea? It must be clarified, it´s not obvious to the reader; this was not mentioned in section 2.2.1.
Lines 305-306: it was an expected result? How the result is explained?
Lines 347-348: Do chickpeas contain luteolin? what is the reason to mention it?
Many similar improvements must be conducted throughout the entire section.
Tables
Table 1- it must include statistical comparison vs NC. SE: include only two decimals. Include a space between numbers and symbols (34.2 + 0.32).
Figures
The figures are very blurred (low resolution) and it is difficult to read the texts and to see the results, for example, Figure 2D
It seems that in Figure 1A only 3 groups were included. Where are the other groups? Are overlapped? Which are the units of this figure?
Conclusions
This section includes information that can´t be concluded from the results of this paper. For example, apoptosis and cell cycle were not evaluated, but evaluating some genes that evolved into these processes is not enough to conclude it. How can it be proved that the results are due to a synergic effect? Concussions must be supported by the results.
In general, the document contains many typos, grammar, and language errors
Reviewer 3 Report
The manuscript presented for review concerns the effect of chickpeas used in the diet on the development of chemically induced colon cancer in mice. The work is well written, although in my opinion it is too long. Authors should shorten the text both in the introduction and in the discussion of the paper. The results are presented in a clear way. I have the following recommendations for the authors of this manuscript:
1. I propose to shorten the text by removing lines 42-59 and 433-437 and other fragments loosely related to the subject of the work.
2. The abbreviation DAI should be explained in the abstract.
3. In the introduction, I would add a few sentences about the cancerogens used.
Round 2
Reviewer 2 Report
Figure 4 and Figure 1A, must be improved. Conclusions are redundant.
Author Response
Figure 4 and Figure 1A, must be improved.
Response from the authors:
Thank you for your observation; we have made the changes to improve the figures.
Conclusions are redundant
Response from the authors:
We appreciate your observation; we have improved the conclusions.